# From Feasting to Fasting: The Arginine Pathway as a Metabolic Switch in Nitrogen-Deprived *Chlamydomonas reinhardtii*

**DOI:** 10.3390/cells12101379

**Published:** 2023-05-13

**Authors:** Lucca de Filipe Rebocho Monteiro, Laís Albuquerque Giraldi, Flavia Vischi Winck

**Affiliations:** 1Laboratory of Regulatory Systems Biology, Center for Nuclear Energy in Agriculture, University of São Paulo, Piracicaba 13416-000, Brazil; 2Department of Botany, Institute of Biosciences, University of São Paulo, São Paulo 05508-090, Brazil; 3Department of Biochemistry, Institute of Chemistry, University of São Paulo, São Paulo 05508-000, Brazil

**Keywords:** systems biology, regulation, amino acids, biomass, arginine, nitric oxide, meta-analysis, motif elucidation

## Abstract

The metabolism of the model microalgae *Chlamydomonas reinhardtii* under nitrogen deprivation is of special interest due to its resulting increment of triacylglycerols (TAGs), that can be applied in biotechnological applications. However, this same condition impairs cell growth, which may limit the microalgae’s large applications. Several studies have identified significant physiological and molecular changes that occur during the transition from an abundant to a low or absent nitrogen supply, explaining in detail the differences in the proteome, metabolome and transcriptome of the cells that may be responsible for and responsive to this condition. However, there are still some intriguing questions that reside in the core of the regulation of these cellular responses that make this process even more interesting and complex. In this scenario, we reviewed the main metabolic pathways that are involved in the response, mining and exploring, through a reanalysis of omics data from previously published datasets, the commonalities among the responses and unraveling unexplained or non-explored mechanisms of the possible regulatory aspects of the response. Proteomics, metabolomics and transcriptomics data were reanalysed using a common strategy, and an in silico gene promoter motif analysis was performed. Together, these results identified and suggested a strong association between the metabolism of amino acids, especially arginine, glutamate and ornithine pathways to the production of TAGs, via the de novo synthesis of lipids. Furthermore, our analysis and data mining indicate that signalling cascades orchestrated with the indirect participation of phosphorylation, nitrosylation and peroxidation events may be essential to the process. The amino acid pathways and the amount of arginine and ornithine available in the cells, at least transiently during nitrogen deprivation, may be in the core of the post-transcriptional, metabolic regulation of this complex phenomenon. Their further exploration is important to the discovery of novel advances in the understanding of microalgae lipids’ production.

## 1. Introduction

Since the publication of its genome sequence and annotation in 2007, the microalgae *Chlamydomonas reinhardtii* (*C. reinhardtii*) has been an important model species for molecular characterisation studies using omics data, especially transcriptome, proteome and metabolome. Over the last decade, these studies revealed several nuances of the *C. reinhardtii* metabolism and aspects of importance for biotechnological applications, such as lipids’ accumulation and biomass production [1,2,3,4,5,6,7,8]. Under different nutritional stress conditions (e.g., deprivation of nitrogen, phosphorus, sulphur, iron and high salinity), *C. reinhardtii* cells’ growth decreases significantly while neutral lipids’ (triacylglycerols—TAGs) content rises inside the cells [2,4]. However, the core cellular mechanisms and biological networks (metabolic and regulatory) that drive, control and link these cellular output phenotypes are not completely understood.

The nitrogen (N) deprivation condition (-N) is one of the well-studied nutritional stressor conditions in *C. reinhardtii* since it is associated with the highest TAGs accumulation [1,2,3,4,5,6,7,8] and induces gametogenesis by activating several mating type-specific genes [9,10]. Both -N-induced processes allow *C. reinhardtii* cells to thrive under unfavourable environmental conditions. Sexual reproduction leads to the development of stress-resistant spores and, due to genetic recombination, may allow for high adaptability in a changing environment (deeply reviewed in [11]). As for the accumulation of lipids, its roles range from a carbon and energy storage and electron sink against oxidative stress [12] to potentially affecting cell positioning in the water column (and, consequently, nutrient and light scavenging), as the occurrence of intracellular lipid droplets affects floatability [13]. Interestingly, a recurrent molecular phenotype under N limitation conditions has been observed, which is the upregulation of the transcripts and proteins of arginine (Arg) biosynthesis and adjacent pathways [1,2,5,7,8,14]. Different hypotheses were stated, such as that Arg functions as an N storage due to its high N to C ratio compared to other amino acids [3,6,7]; or that proteins of these metabolic Arg-related pathways have the role of quickly capturing any N that becomes available (i.e., priming function) [5]; and finally, Arg may be a precursor in the biosynthesis of signalling molecules such as polyamines or nitric oxide (NO), as verified previously in *Arabidopsis* under biotic and abiotic stresses [15]. Besides this diverse and conflicting information about the physiological functions of the Arg metabolism in stress-related responses, there is only Incipient and suggestive information on the regulatory and signalling implications of the Arg metabolism in *C. reinhardtii.*

Such complex responses are most probably controlled by multi-factorial responses, which can be related to the characteristic of multiple simultaneous intermediate outcomes derived from the regulation of a common metabolic pathway. A possible way to identify such molecular circumstances is to analyse the co-regulated genes or common patterns of co-activated regulatory genomic elements. Thus, the identification of sequence motifs shared by the promoter regions of a set of co-expressed genes is commonly used for seeking regulatory elements associated with the transcriptional mechanism of a given characteristic [16]. This approach may identify patterns of co-regulation between co-expressed genes and reveal unexpected associations between the selected genes and metabolic or regulatory pathways [17,18]. So far, there are few studies on *C. reinhardtii* based on a conserved promoter motif analysis and focused on seeking the identification of new regulators of specific physiological processes. Some examples include analyses of the Carbon Concentrating Mechanism [19,20], control of zygotic gene transcription [21] and response to phosphorus starvation [22]. Therefore, promoter motif analysis tools could aid the identification of regulators and pathways associated with Arg metabolism and nitrogen assimilation, in the context of the interface between cell growth quiescence and the accumulation of TAGs during -N.

To better understand the Arg metabolism in *C. reinhardtii* under -N, and its possible role in the regulation of TAGs accumulation and growth arrest, we aimed to reanalyse published omics (proteomics, metabolomics and transcriptomics) datasets and to identify conserved promoter sequence DNA motifs using in silico tools. The analysis of the conserved gene promoter regions from the Arg metabolism’s co-expressed genes during -N and the identification of the population of other genes that share the same conserved gene promoter patterns could indicate possible novel components of the regulatory subnetworks of -N, that may play an important role in the regulation of TAGs accumulation.

The high number of published omics (proteomics, metabolomics and transcriptomics) datasets of -N responses in *C. reinhardtii* reported a valuable resource to search for novel shared promoter DNA motifs on the co-expressed genes using in silico tools. Therefore, we aimed to better determine the role of the amino acid metabolism (Arg-related in particular) and other related metabolic pathways of the central metabolism and TAGs biosynthesis from a regulatory point of view. With the target reanalysis of published omics data, we sought to assess the contribution of the amino acid catabolism to TAG biosynthesis, as amino acids may provide C skeletons to the de novo fatty acid anabolism, and how this response may be generated and associated with the regulation of other genes related to -N responses. Complementarily, the functional enrichment analysis of gene clusters sharing conserved promoter DNA motifs may shed light on novel metabolic and regulatory modules integrating amino acid catabolism with TAGs accumulation, hinting at potential novel regulators and signalling events related to TAGs biosynthesis under -N. 

## 2. Materials and Methods

### 2.1. Chlamydomonas Strains of Reanalysed Data

The Chlamydomonas cells investigated in the three main previous studies reanalysed in this work are from the same genetic background 137c; however, not all of them are exactly from the same strain of the Chlamydomonas genome sequencing project. The work of Park et al. (2015) [5] was performed with the cell wall-deficient mutant of *C. reinhardtii* CC-400 cw15 mt+, obtained from the Chlamydomonas Resource Center at the University of Minnesota (St. Paul, MN, USA). This strain is a cell wall mutant with a very small amount or absence of a cell wall, which requires minor amounts of acetate (0.2%) to survive in liquid cultures. Wase and colleagues performed the experiments with the *C. reinhardtii* strain CC125 from the Chlamydomonas Genetics Center (http://www.chlamy.org/ (accessed on 05 march 2023)); the wild type 137c strain of Chlamydomonas served as the background lineage for the generation of the strain CC503 cw92 mt+. Valledor and colleagues [7] performed their systems analysis using the *C. reinhardtii* CC-503 cw92, mt+, agg1+, nit1, nit2 (available at the Chlamydomonas Culture Collection, Duke University) strain, which was the same strain used for the Chlamydomonas genome sequencing project [23]. NNG-induced mutants CC400 cw15 and CC503 cw92 were both generated from the *C. reinhardtii* linage wild type 137c (CC125), originally obtained from Prof. R. P. Levine by Hyam and Davies, who generated the mutant cw92 lines, which was the same strain used for the Chlamydomonas genome sequence and annotation project [24]. Both cells are mutants of cell wall production, showing minute quantities or an absence of the cell wall. The strains CC503 cw92 mt+, CC400 cw15 mt+ and the CC125 have also been shown to have specific mutations, such as *nit1* (nitrate reductase gene) and *nit2* (transcriptional regulator of nitrogen assimilation and nitrate reductase gene) mutations, which implies that cells cannot use nitrate as their sole nitrogen source but require nitrate intracellularly for surviving [25]. Goodenough and colleagues [14] and Blaby and colleagues [1] studied the CC4349 cells, which are a cw15 (nit1 NIT2 mt−) cell wall-less mutant strain, with non-arginine-requiring dependency, capable of proliferating under phototrophic conditions.

### 2.2. Omics Data Mining: Dataset Reanalysis and Functional Annotation

The proteome and metabolome raw data from three main previous studies [5,7,8] that described the cellular responses to N deprivation in *C. reinhardtii* were retrieved (Appendix A) and reanalysed by recalculating the fold-change values for the proteins and metabolites reported based on the Log2 values of quantification measurements extracted from the raw data published.

We determined the fold-change for the proteins and metabolites reported previously by calculating the differentially expressed (DE) proteins or metabolites at the time point 24 h -N relative to the time point 0 h, during which cells were still under N-replete conditions in each study (Appendix A) by Park et al. [5] and Valledor et al. [7], or by calculating their logarithm fold-change (Log2FC) ratios from 24 h under -N relative to N-replete conditions (Appendix A), as in the study by Wase et al. [8]. This analysis was performed to assure that the comparative analysis would be performed comparing the same similar conditions. The calculated Log2FC values above and below zero were summarised as “upregulations” and “downregulations”, respectively (Appendix A). We, therefore, selected a group of DE proteins and metabolites related to the main metabolic pathways, which were subsequently analysed for co-expression at 24 h -N. These proteins and metabolites were selected from the following metabolic maps from the KEGG PATHWAY database [26,27]: cre00220 (Arg biosynthesis), cre00330 (Arg and proline metabolism), cre00250 (alanine, aspartate and glutamate metabolism), cre00910 (N metabolism), cre00710 (carbon fixation in photosynthetic organisms), cre00010 (glycolysis/gluconeogenesis), cre00020 (TCA cycle), cre00061 (fatty acid biosynthesis), cre00630 (glyoxylate and dicarboxylate metabolism), cre00260 (glycine, serine and threonine metabolism), cre00270 (cysteine and methionine metabolism), cre00280 (valine, leucine and isoleucine degradation), cre00350 (tyrosine metabolism), cre00360 (phenylalanine metabolism) and cre00410 (ꞵ-alanine metabolism). Additional pathways’ data were retrieved from the PMN Chlamycyc 8.0 database [21,28] and from Li-beisson et al. [29]. Phytozome (v5.6 annotation) [30] and Uniprot [31] databases were also consulted for gene/enzyme names. The list containing all proteins and metabolites retrieved from the databases can be found at Appendix A. To complement the protein expression data, transcriptome data from three other previous studies [1,2,14] were reanalysed using an in-house developed platform, named the Phycomine data warehouse (unpublished data) [32], exclusively considering CC-4349 [1,14] or CC-2137 [2] strains; Z-score thresholds between −1.5 and 1.5, respectively; and the transcriptome of cells at the time point 24 h under -N relative to unstressed, control conditions. This time point was chosen for the reanalysis since it is characterised by an increased TAGs relative abundance and markedly decreased cell growth rates, compared to unstressed conditions [2,5,7].

### 2.3. Promoter Motif Analysis and In Silico Gene Target Association Inference

*C. reinhardtii* DNA sequences 1500 bp upstream of each transcription starting site (TSS) for all annotated genes were retrieved from the genome sequence (annotation v.5.6) (*C. reinhardtii* v5.6, Phytozome genome ID: 281, NCBI taxonomy ID: 3055, Accession ID:ABCN02000000) and trimmed if a neighbouring gene was found within the vicinity of this region. The putative promoter regions (up to 1500 bp) were retrieved, and the promoter region of the differentially expressed genes was retrieved and common DNA motifs were queried using the MEME tool (Appendix A). The maximum sequence length of the putative region selected as gene promoters was defined based on previous evidence of the position of the most likely CIS elements already described experimentally for Chlamydomonas [20].

Three groups of promoter sequences were organised: (1) gene promoters from Arg metabolism-related genes encoding upregulated proteins (Appendix A); (2) gene promoters from Arg and central C metabolism genes (amino acid degradation, glycolysis, fatty acid synthesis and TAG anabolism pathways) encoding upregulated proteins (Appendix A); (3) gene promoters from Arg metabolism and central C metabolism genes encoding upregulated proteins and promoters from all fatty acid synthesis genes encoding up- or downregulated proteins (Appendix A). Conserved DNA sequence motifs on the gene promoters were queried using the MEME motif discovery tool [33] using default parameters, and anr (any number of repeats) settings and a significance E-value threshold of 0.05. Only the two most significant motifs obtained from each dataset were further investigated (M1–M2, Appendix A).

The common DNA motifs (Appendix A) from each group of promoter sequences were functionally annotated. Each conserved motif was queried against the *C. reinhardtii* promoterome (Appendix A) using the MAST motif analysis tool [33], in order to associate each motif with a cluster of possible target genes possessing the conserved motif in their promoter region (M1–M6, Appendix A, respectively). The obtained clusters were subjected to a Gene Ontology (GO) enrichment analysis of biological process (BP) terms, using the BinGO plugin within Cytoscape [27] with a hypergeometric test, Benjamini and Hochberg false discovery rate (FDR) correction and a significance level cut-off of 0.05. Only the overrepresented categories were selected to be analysed further.

## 3. Results

We revisited and reanalysed published omics data (proteomics, metabolomics and transcriptomics data) and performed a gene promoter motif in silico analysis on target gene subsets. Our meta-analysis of omics data reinforced the knowledge of the effects of -N into the central metabolism (C:N balance) and suggested a significant role of amino acids’ synthesis and degradation and related metabolic pathways, Arg-related in particular, as possible regulatory nodes of the stress response leading to TAGs accumulation and cell growth arrest. Subsequently, we identified the conserved common promoter DNA sequence motifs from co-expressed genes, giving us a further understanding of the regulatory mechanisms of target pathways at the transcriptional level.

### 3.1. Metabolome, Proteome and Transcriptome Reanalysis Indicates Protagonist Amino Acids in Nitrogen Starvation Response

Most amino acids had their relative abundance diminished in response to -N, while the enzymes of the Arg biosynthesis increased. Though arginine, asparagine, glutamine and ornithine (Figure 1), together with lysine and tryptophan, were decreased, this pattern was not restricted to amino acids containing N in their side chains (Appendix A). Alanine, aspartate, glutamate and proline were also decreased. From the 22 proteinogenic and non-proteinogenic amino acids analysed, only ꞵ-alanine, L-homoserine and pyroglutamic acid were increased [8]. Strikingly, despite -N, key enzymes in the Arg biosynthesis from glutamate (GSN1, GLN1, CMP2 and AGS1) were upregulated in all three datasets (Figure 1). Although its N-containing intermediates were decreased, most enzymes from this pathway were increased, along with the ammonium transporter AMT4, nitrite reductase (NII1) and ammonia-consuming glutamine biosynthesis enzymes (Figure 1). Protein expression data for nitrate reductase (Cre09.g410950, Cre06.g303050), NAR transporters (Cre07.g335600, Cre06.g309000, Cre01.g012050, Cre04.g217915, Cre09.g410900, Cre12.g541250) and NRT transporters (Cre03.g150151, Cre03.g150101, Cre02.g110800, Cre09.g410800, Cre09.g396000, Cre09.g410850, Cre04.g224700) were not found in the reanalysed datasets.

Several amino acid catabolic enzymes were upregulated in one of the investigated datasets during the 24 h under the -N condition [5] (Appendix A). The upregulated aspartate aminotransferases ASAT1 [7], ASAT3 [5] and ASAT4 [5] (Figure 1) participated in pyruvate-forming proline and cysteine catabolism, phenylalanine and tyrosine metabolism and glutamate synthesis from 2-oxoglutarate. The amine oxidases AMX1 [5,7] and AMX2 [5] were also upregulated, having roles in the ꞵ-alanine biosynthesis and phenylalanine, tyrosine and glycine (methylglyoxal-forming) catabolism. The same pattern of expression was observed with the Cre03.g181200 and Cre09.g399030 enzymes [5], with the first having roles in leucine degradation to acetyl-CoA, and the latter being involved with phenylalanine degradation. Still, OTA1, a key enzyme in isoleucine degradation to acetyl-CoA, was downregulated in all three late -N datasets, together with SGA1, AGT1 AAT1 alanine and serine/glycine degradation enzymes [7] (Figure 1). As LAO1, a key participant in L-amino acid catabolism and ammonia recycling, was upregulated in most datasets [5,8], we can conclude that amino acid degradation was overall upregulated.

Due to a lack of protein expression data concerning Arg catabolism pathways, three transcriptomics datasets [1,2,14] were also reanalysed for the 24 h -N condition (see Section 2). Two co-expression gene clusters were identified, encompassing early responses -N (0–2 h) and late responses (12–48 h) time points at the -N condition (Appendix A). Most 24 h -N transcripts had normalised expression data between −1.5 and 1.5 (Z-score values), particularly within the Goodenough et al. [14] dataset. The putative upregulation of *ODC1* [2], *ODC2* [1] and *CPA1* [1] during this time point, with Z-scores above 1.5, is suggestive of Arg degradation to putrescine. Interestingly, concerning the 0–2 h -N period, the *AMI2*, *AIH1* and *AOF6* putative upregulation verified in most of the datasets (Appendix A) might indicate Arg degradation to putrescine and GABA, although *ODC1* and *ALD5* were downregulated in two of them. Together with *AOF6*, the amine oxidases *AMX1* and *AMX2* were also upregulated in this period, with the latter in all early -N datasets. Transcripts encoding the PII protein (*GLB1*), a key activator of the Arg biosynthesis pathway in plants [34,35,36] modulated by NO and -N [35,36], were putatively upregulated in the three datasets during early and late time points under the -N condition (Appendix A) [5].

Furthermore, the expression patterns of proteins and transcripts from central C metabolism pathways were not homogenous. For instance, the TCA cycle was upregulated, as underlined by CIS1, ACLB1, ACH1, IDHs, SDH2 and MDNs’ upregulation in most datasets, followed by glyoxylate cycle downregulation (Figure 2). The Calvin cycle and gluconeogenesis were likely downregulated, as evidenced by the downregulation of the fructose-1,6-bisphosphatase (FBP2), a key participant in both pathways, and Rubisco small subunit 2 (RBCS2) (Figure 2). Though DLA2 was downregulated in two datasets [5,7] (Figure 2) and the expression data concerning upstream glycolytic steps were sparse, the pyruvate biosynthesis from glycolysis and its degradation to acetyl-CoA were upregulated, together with the TAG anabolism pathway from fatty acids and glycerol. More notably, despite this pattern of expression followed by TAG anabolism, acetyl-CoA carboxylation to malonyl-CoA was downregulated, as indicated by the downregulation of BCC1 and BCX1 forming less malonyl-CoA, while the downstream step of the de novo fatty acid biosynthesis pathway forming palmitoyl-CoA biosynthesis was found to be upregulated in all datasets (Figure 2). In summary, the proteomics data reviewed may establish a route to acetyl-CoA accumulation during 24 h -N but fail to ascertain its destination precisely. This issue can be extended to carbon (C) assimilation by the cell, as the Calvin cycle is in general downregulated, but omics data concerning acetate assimilation are inconsistent. Therefore, the reanalysis of the omics data indicates that the chloroplast function is affected by the resulting downregulation of several enzymes from the Calvin cycle, which is a similar response found in vascular plants to overcome the excess reducing power and oxidative stress generated in response to abiotic stressors. However, how it is precisely connected to the TAGs biosynthesis is still not fully clarified.

During the -N response, the cells naturally produce reactive oxidising species, and the overexpression of the enzymes of the TCA cycle are likely related to the regeneration of oxidised forms of co-enzymes for reducing the impact of the high concentration of reducing power (via reduced co-enzymes) accumulated in the cells under abiotic stress responses.

We investigated the transcriptional patterns of genes related to fatty acid (de novo) and TAG biosynthesis, acetate assimilation and acetyl-CoA synthesis from pyruvate through the reanalysis of three transcriptomics datasets [1,2,14], considering the 24 h -N time point (see Section 2). As indicated by Appendix A, different clusters of genes related to TAG and, remarkably, de novo fatty acid biosynthesis were putatively upregulated at 24 h -N. While most of the queried genes were upregulated within the Blaby et al. [1] dataset (Appendix A), with normalised expression values (Z-scores) above 1.5, different (and fewer) genes were upregulated in Boyle et al. [2] (Appendix A), by including key TAG biosynthesis (*PDAT1*) and de novo fatty acid synthesis (*MCT2*, *KAS3*) enzymes. None of the analysed genes scored below -1.5 or above 1.5 in the Goodenough et al. [14] dataset (Appendix A), and the same was observed for queried acetate assimilation (Appendix A) and acetyl-CoA synthesis (Appendix A) genes. Concerning acetate assimilation enzymes, only acetate kinase (Cre09.g396700) was putatively upregulated [2] (Appendix A), as other transcripts had normalised expression values (Z-scores) between −1.5 and 1.5 (Appendix A) in the -N condition. Another acetate kinase (Cre17.g709850) was also upregulated in Park et al. [5] at late time points for the -N proteomics dataset (Appendix A). Although expression data from transcripts associated with the acetyl-CoA synthesis from pyruvate were similarly scarce, the putative upregulation of *PDC2*, *PDH2* [1] (Appendix A) and *DLA1* [2] (Appendix A) key enzymes was detected. In summary, the transcriptomics datasets suggest the increase in pyruvate concentration, with a high production of acetyl-CoA (likely also from citrate) and the induction of the fatty acids (de novo), and TAG biosynthesis, indicating that not only are membrane lipids remobilised during the stress response, but new TAGs are synthesised in the cells.

### 3.2. Promoter Motif Analysis Suggests Novel Regulatory Modules

To ascertain if the co-expression patterns observed in the omics data reanalysis were associated with the transcriptional co-regulation of gene expression, common DNA motifs of the putative promoter regions of genes encoding the upregulated proteins listed in Appendix A were obtained and analysed through multiple bioinformatic tools as previously described (see Section 2). Six statistically conserved significant promoter motifs (M1–M6) were identified through a motif analysis using MEME suite (Figure 3A, Appendix A). Motifs M1–M2 were obtained by using putative promoters from Arg metabolism genes as input (Appendix A). Motifs M3–M4 were inferred from the putative promoters from Arg metabolism plus central C metabolism genes (Appendix A), with the latter encompassing amino acid degradation, glycolysis, TCA, fatty acid synthesis and TAG anabolism pathways. Motifs M5–M6 were identified using the same pathways from M3–M4 as the input, though considering all fatty acid synthesis genes from Appendix A, regardless of their protein up- or downregulation status.

Furthermore, we searched for conserved motifs in the promoterome of *C. reinhardtii* using the MAST tool (Appendix A), resulting in the clusters shown in Figure 3A (Appendix A). Eight genes were found to possess all M1–M6 motifs within their promoter regions, with five of them being *CHLAMYDOMONAS-SPECIFIC FAMILY U PROTEIN* genes (Cre02.g142947, Cre04.g217977, Cre05.g235228, Cre11.g467621 and Cre31.g758447), predicted to encode the coiled-coil domain-bearing uncharacterised protein family in *C. reinhardtii*. The genes Cre02.g142947, Cre05.g235228 and Cre11.g467621 are suggested to be part of the RNA-BINDING ASCH DOMAIN PROTEIN and Xylogalacturonan beta-1,3-xylosyltransferase/Xylogalacturonan xylosyltransferase gene families (Phytozome IDs 123395446 and 124077440, respectively). Cre02.g075400 and Cre16.g685451, another two of the genes found to bear all M1–M6 promoter motifs, respectively encode a PLAC8 motif-containing protein (IPR006461), exhibiting a thioredoxin-like fold domain (IPR012336), and a protein containing an enolase N-terminal domain-like domain (IPR029017). A functional analysis of Gene Ontology overrepresented terms for the M1–M6 target gene clusters through BinGO (Appendix A) revealed that GO biological processes (BP) terms related to arginine, glutamine and glutamate metabolism pathways were significantly enriched in the genes found containing the M1 motif, together with the “protein phosphorylation” BP term (Figure 3B, Appendix A). The M6 cluster was also found to be significantly associated with the latter term, as 32 of its genes were linked to “protein phosphorylation” (Appendix A). Still, clusters M2–M5 did not present any significant functional enrichments for BP terms (Appendix A).

The promoter motif, gene clustering and cluster functional enrichment analyses performed provided novel insights into the regulatory landscape of the 24 h -N condition in *C. reinhardtii*. The retrieved conserved promoter motifs could harbour binding sites for transcription factors, thus contributing to the patterns of expression observed in the reanalysis performed previously.

## 4. Discussion

Our results of the reanalyses of the selected metabolome, proteome and transcriptome datasets indicate that priming of the Arg metabolism pathways occurs at late -N stages (24 h on), and it may exert an important role inducing TAGs accumulation, likely functioning as a metabolic switch. The results of our gene promoter motif analysis also indicate the presence of regulatory modules integrating the transcriptional regulation of TAGs accumulation with N assimilation.

### 4.1. General Role of Arg Metabolism as Priming and Metabolic Switch in Nitrogen Deprivation

During the late stage of -N (24 h), N uptake and N assimilation into glutamine and glutamate synthesis from glutamine were increased (Figure 1), and the enzymes from the Arg biosynthesis and catabolism pathways were upregulated (Figure 1 and Appendix A). This was also documented in previous studies of genomic-based metabolic reconstruction [37] together with the upregulation of polyamine transporters, hinting at an attempt to scavenge any intra- or extracellular N available [5,37].

The mechanisms of nitrogen scavenge and the uptake of ammonium, nitrite and nitrate in Chlamydomonas have been previously described, revealing transporters and co-regulated genes. The overexpression of nitrite transporters at the plasma membrane (NAR1.3, NAR1.4, NAR1.6) and three in the chloroplast (NAR1.1, NAR1.2, NAR1.5) under low nitrate concentration or -N has been associated with the expression of genes related to the nitrogen assimilation [38].

It is important to note that for nitrate and nitrite uptake and related signalling under -N, the regulator NIT2 appears as essential for the control of the expression of nitrate-assimilation genes and transporters, which are negatively regulated by the presence of ammonium in the media and positively regulated by nitrate and -N. The function of NIT2, a transcription factor protein, has been linked to the transcriptional regulation of several target genes, including the nitrate reductase (NIA1) gene [39], which may be in the core of the transcriptional control of the -N response. However, this function requires intracellular nitrate and might also be negatively regulated by other compounds [40]. In fact, we did not find NIT2 gene expression in the cells investigated in this present study, as expected, since the cells have a *nit2* mutation. Consequently, NIA1 (nitrate reductase) is also absent. The function of NIT2 has been related to the presence of NO, which might induce NIR expression, leading to the reduction of nitrate and allowing its assimilation in an adjustable manner, while NO can inhibit the high-affinity transport of nitrate, nitrite and ammonium in a post-transcriptional process, as NO can also reduce the activity of NR (nitrate reductase) [39,40,41]. Moreover, NOFNiR (NO-forming nitrite reductase) has also been attributed to generating NO, which inhibits NRT2 and NIA1 expression [39].

While nitrate transporters and the NIT2 regulator have been described as important elements of the signalling of nitrate-assimilation genes, the transport of ammonium (the nitrogen source used in the studies revised here) is performed by membrane transporters, including AMT4. These transporters are highly expressed under -N and sole arginine-fed cultures of Chlamydomonas, either in mixotrophic or phototrophic conditions. The difference is that under mixotrophic growth and arginine as the sole N source, the cells show similar modulation of the expression of genes responsible for nitrate assimilation and responsive to -N, but they do not show cell growth arrest as observed in the -N condition, indicating that acetate is related to the nitrogen metabolism under -N. Of note, in contrast to mixotrophy, the cells under phototrophic growth treated with the arginine supply as the sole N source show a reduced cell growth rate [41].

That may be the reason why the cells metabolically respond to ammonium deprivation upregulating the ARG9 and GSN1 even in the absence of NIT2, the known master regulator of nitrogen assimilation. Therefore, our results suggest the presence of a multi-combinatorial regulation of nitrogen assimilation with a set of common genes responsive to nitrate and ammonium deprivation.

The cellular response to -N has also been associated with the upregulation and induction of LAO1 (L-amino acid oxidase). This enzyme is a periplasmic protein in Chlamydomonas, and it is responsible for the deamination of most amino acids. As it is upregulated under -N conditions, its function is importantly related to provide the cells with the incorporation of a nitrogen source (in this case, ammonia from the deaminated amino acids). Once free, ammonia is uptaken from the environment, allowing cells to survive using most amino acids as the only source of N under the mixotrophic condition. In long-term cell growth (12 days) on sole L-amino acids as the nitrogen source, wild type cells (CC5325) have been shown to survive using 19 out of the 20 proteinogenic amino acids, and LAO1 appears as essential to the efficient deamination of 14 of them in short-term growth (4 days) and its function seems to be improved or activated over time for some specific amino acids (L-histidine, L-aspartic acid, L-glutamic acid, L-threonine and glycine) [42]. However, in the omics data reanalysis performed here, the cells were deprived from nitrogen and have no exterior amino acid source; therefore, they had induced the overexpression of LAO1 but no N assimilation from external sources.

The omics reanalysis also suggests that the reported upregulation of the TCA cycle, glycolysis and amino acid degradation pathways shuttles C skeletons to acetyl-CoA synthesis, bolstering de novo fatty acid synthesis and perhaps connoting the co-occurrence of regulatory effects. Interestingly, although a decrease in the N-rich amino acid pool was observed (Appendix A) together with protein anabolism hindrance and increased proteolysis [5], indicating N-sparing processes [6,43], a decrease in the levels of amino acids without N in their side chains occurred as well (Appendix A). This general reduction in amino acids’ content alongside the upregulation of enzymes from glycolytic pyruvate synthesis and its low conversion to Acetyl-CoA via the reduced expression of DLA2, and the simultaneous overexpression of enzymes and accumulation of metabolites from the TCA cycle (Figure 2), reinforce the indication that carbons from amino acids may be redirected for TAGs accumulation from citrate conversion to acetyl-CoA via CIS1 and ACLB1 overexpression, contributing to de novo fatty acid synthesis [7,8,44,45], which may also indirectly connected to starch catabolism [5]. As has been well-reported by Liang et al. [46], the catabolism of branched-chain amino acids (valine, leucine, isoleucine), in all the metabolomics datasets we analysed (Appendix A), may provide C precursors (acetyl-CoAs) for fatty acid biosynthesis and could have signalling roles by influencing the cellular C:N balance [47]. This consideration has interesting ramifications, as nutrient availability-derived changes in C:N ratios can induce lipids accumulation in microalgae [48] and reshuffle TCA and N assimilation [49]. Furthermore, the role of key C:N content regulators has been more thoroughly investigated in recent years; one of these, TOR (target of rapamycin), orchestrates *Chlamydomonas* central C pathways [50,51] and, in particular, amino acid metabolism [52]. Moreover, both the cellular Arg sensor for mTORC1 (CASTOR1) found in humans and some *Arabidopsis* putative amino acid sensors share regulatory ACT domains and ACT conservation patterns, hinting at a common origin [53].

These observations indicate that several mechanisms are participating in the regulation of TAGs accumulation, with regulatory events occurring at the transcriptional and post-transcriptional levels.

### 4.2. Integrative Regulation of Transcriptional Control of N Assimilation and Arginine Metabolism

The relevance of the metabolic pathways affected by -N, especially the one related to the arginine metabolism, draws our attention to the likely existence of a regulatory network that might control the cellular responses inducing the cell growth arrest and TAGs accumulation under the -N condition. This hypothetical regulatory network and its network modules would be composed of genes containing common promoter DNA sequence motif(s), which may function as *Cis* regulatory elements recognisable by one or more transcriptional regulatory proteins.

Our results on the gene promoter motif analysis of the upregulated genes related to the Arg metabolism revealed DNA sequence motifs (M1, M3 and M5), highly conserved in their promoter region and in the promoter region of other genes, that may be associated with the function of this metabolic pathway (Appendix A).

The promoter motif analysis showed the simultaneous presence of both M1, M3 and M5 motifs in the promoter region of only 68 genes (among differentially expressed and non-differentially expressed genes), including NIT2 (Cre03.g177700), Acetylornithine aminotransferase (Cre06.g278163) and N-acetyl glutamate synthase (Cre16.g694850) to mention a few (please see Supp. Material 12). The presence of this combination of motifs in the promoter of these genes suggests that they may be functionally associated, and that a common transcriptional regulator(s) might co-regulate their expression, suggesting that transcriptional control through Nit2 is transcriptionally integrated with the regulation of the Arg metabolism.

Furthermore, other amino acid transporters and permeases might also play a role in the -N response. For instance, the sequence motifs M3 and M5 were found in the gene promoters of Nit2 (Cre03.g177700), Nrt2.4, a nitrite/nitrate transporter (Cre03.g150101), in the promoter region of one amino acid transporter (Cre02.g145750) and two amino acids’ permeases Bat1 (Cre01.g023650 and Cre07.g348040). These two motifs have also been found in the promoter of the 26 s proteasome regulator subunit (Cre06.g278256), Mapk6 (Cre12.g508900) and Cytochrome b6f (Cre12.g546150), among others (Suppl. Material 12). These findings also raise some questions if the cells can activate other amino acids transporters and permeases in conditions other than -N, and that nitrogen uptake and protein degradation may be commonly regulated by the same set of regulators.

Therefore, as no Arg nor ornithine accumulation has been reported for this late -N condition (Figure 1) [34,43], our analysis adds further validation to the hypothesis that the upregulation of the Arg pathways during late -N serves as a means to anticipate, rapidly acquire and assimilate exogenous N after stress alleviation [5,7,43,54].

Of note, our results showed five proteins containing all six M1–M6 conserved promoter motifs in their genes which belong to the protein family *CHLAMYDOMONAS-SPECIFIC FAMILY U PROTEIN* (Cre02.g142947, Cre04.g217977, Cre05.g235228, Cre11.g467621 and Cre31.g758447). This protein family may contain the PUA (PseudoUridine synthase and Archaeosine transglycosylase) domain and a highly conserved RNA-binding motif, likely involved in post-transcriptional modification and ribosome degradation [55]. This group of proteins may be involved with the reduced total amount of proteins observed in *C. reinhardtii* cells under -N, likely regulating protein degradation together with the proteasome machinery.

Our omics data reanalysis indicates that the allocation of C is a process widely layered by the post-transcriptional regulation in *C. reinhardtii*. However, we recognise that our analysis is limited because it is mostly based on transcriptomics data. Moreover, the proteomic (Figure 2) and transcriptomic (Appendix A) data regarding the de novo fatty acid biosynthesis during 24 h -N are strikingly contrasting, especially related to the destination of pyruvate and acetyl-CoA, suggesting that a still-unknown path or complex regulatory process involving signalling events is responsible for the reported patterns of the expression and destination of metabolic precursors [5].

Recently, alternative splicing events have been found to be associated with the function of the citrate cycle, fatty acid metabolism and the remodelling of membrane lipids during -N in *C. reinhardtii* [56]. This is further evidence that the de novo synthesis is not the sole source of FAs during TAG synthesis [5,29]. Conversely, -N-induced microRNAs have been suggested to regulate lipid metabolism [57]. More importantly, enzymes involved with fatty acid synthesis are prone to regulation through phosphorylation in -N conditions [58], together with rapidly responsive reversible cysteine thiol oxidation [59]. Likewise, these mechanisms may explain the inconsistencies found in acetate uptake expression patterns (Figure 2 and Appendix A). With the stress-induced impairment of photosynthesis (Figure 2) and reduced cell growth, exogenous acetate not only serves as an alternative organic C source and substrate for fatty acid synthesis [14,60,61,62] but also modulates TAGs accumulation [14,60]. In this sense, acetate uptake and further metabolisation pathways could be prone to multi-level regulation as well.

### 4.3. Insights of the Regulation and Induction of Signalling Mechanisms

Although this reanalysis mainly investigated the metabolic state of the cell during late -N, transcriptomics data concerning early responses at shorter time points (0–2 h -N) brought insightful expression patterns. The upregulation of putrescine and GABA biosynthesis during early -N (Appendix A) suggests that these compounds act as key regulators of the transition to a stressed state, as polyamines modulate stress responses in microalgae (reviewed in [63]) and cell growth in *C. reinhardtii* [64,65]. As previously reported, GABA, a major metabolic link between C and N metabolism [66], takes part in plant growth (reviewed in [66,67]), energy metabolism [66] and C:N ratio [66,67,68,69] regulation. Similarly, the early upregulation of hydrogen peroxide-producing amine oxidases *AMX1*, *AMX2* and *AOF6* (Appendix A) in response to -N indicates that H_2_O_2_ will also have signalling roles in the relay of a state of stress throughout the cell. H_2_O_2_, a known stress signal transducer in plants [70,71,72], has been shown to induce autophagy [73,74] and early changes in *C. reinhardtii* transcriptome, including the downregulation of the central C metabolism [75], and to modulate its cell cycle together with NO-related events [76,77].

NO production in Chlamydomonas has been related to the degradation of cytochrome b6f in cells under -N, a process that has been shown to be regulated by the nitrate reductase (NIT1) enzyme, which requires the function of nitrite reductase (NAR1) to modulate the intracellular availability of NO and ammonium produced from the nitrite uptake. However, the nitrate reductase function seems to be dispensable for NO production in cells under -N conditions [78]. In our results, the conserved promoter motifs M3 and M5 found in the cytochrome b6f gene were also found conserved in the promoter of the genes encoding the protein required for the cytochrome b6f assembly (Cre12.g537850), Glutathione s-transferase (Cre12.g508850), MAPK6 (Cre12.g508900) and 26S proteasome regulator subunit (Cre06.g278256), among other genes. This may indicate that cytochrome b6f degradation may also be influenced by signalling events and the redox cellular state.

Furthermore, NO is also known as a crucial regulator of *C. reinhardtii* stress responses (reviewed in [79]), and it is intimately related to the Arg and polyamine metabolisms [34,36,80]. This gaseous messenger has widespread roles in the control of the N metabolism, acting as a negative signal to nitrate assimilation in favour of ammonium assimilation [39,81,82,83].

Therefore, the omics reanalysis we performed indicates that NO might be produced in the cells by another pathway and function as a second messenger of the TAGs biosynthesis in cells under mixotrophy and -N conditions, considering that cells have one or more nitrate reductases that are non-expressed or non-induced, with a reduced capacity to generate NO via Nit1 [84]. Furthermore, the motif M2 was found conserved in the promoter of genes from urea transporter DUR31 (Cre08.g360250) and nitrite reductase NII1 (Cre09.g410750), among others, but another motif (M1) was found conserved in the promoter of Nit2 (Cre03.g177700), suggesting that different regulatory modules may be controlled by a different set of transcriptional regulators affecting nitrogen assimilation in *C. reinhardtii*.

Moreover, NO transcriptionally inhibits the PII chloroplast signal transduction protein (*GLB1*) [85], a well-known regulator of the C:N ratio (reviewed in [86]), glutamine, ATP and 2-oxoglutarate sensor [86] and activator of AGK1 [34,87], which is a key enzyme in the Arg biosynthesis pathway. Interestingly, *GLB1* transcripts were upregulated in early -N datasets (Appendix A) and the PII protein was upregulated during late -N (Appendix A) together with AGK1 (Figure 1). This corroborates the finding that -N increases *GLB1* expression [35]; however, the interplay between NO and early -N signalling in *C. reinhardtii* remains poorly understood. Moreover, the PII protein has been found to exert negative control over acetyl-CoA carboxylase activity in *C. reinhardtii* under -N, leading to higher TAG yields. PII may tune down TAG accumulation via the control of fatty acid biosynthesis, possibly by modulating ACCase activity, which is highly expressed in Chlamydomonas. However, the authors also indicate that the depletion of Arg in Arg-dependent strains might additionally increase the TAG synthesis and lipid bodies accumulation via an unknown route [85]. This response could be connected to the observed high abundance of citrate and possible activation of the acyl-CoA-independent pathway for TAGs accumulation in the -N condition.

NO is also recognised by indirectly repressing the function of the NIT2 transcription factor, the known master regulator of nitrate assimilation [40,81], modulating the C metabolism as well [88]. Other evidence of the role of NO in TAGs accumulation points out that enzymes from the *C. reinhardtii* lipid metabolism (such as ACP1) are prone to nitrosylation [89], suggesting that their activity could be modulated by NO. As such, NO—and its interaction with PII in particular—may be an important regulator of the C:N balance during responses to -N in microalgae. This alludes to the possibility of a negative feedback loop mediated by Arg degradation to NO through a putative NO synthase-like enzyme [80,81,83,90], leading to the post-translational control of AGK1, and other enzymes of polyamine pathways in *C. reinhardtii*.

In conclusion, the careful reanalysis of omics data from the studies of -N in *C. reinhardtii* brought new insights into the role of specific amino acids in the metabolic regulation of the TAGs synthesis, indicating that the initial and early responses of the cell metabolism directly affect cell growth through the modulation of the Arg catabolic pathway and consequently polyamine synthesis, suggesting that the transient accumulation of Arg activates a sequence of signalling events that, in the second moment of the response, initiate a series of molecular events that induce TAGs accumulation. This is likely through the effect of NO as a signalling molecule.

Moreover, our omics reanalyses and the identification of conserved promoter motifs suggest the existence of regulatory modules coordinating the transition to TAGs accumulation in late -N, indicating that Arg metabolism and N assimilation are transcriptionally co-regulated with different regulatory modules likely coordinating protein degradation, amino acids transport and catabolism and signalling events. These results suggest that TAGs accumulation can be controlled without interfering with the function of regulatory modules of cell growth, and this brings hope for improving the further biotechnological large-scale production of microalgae lipids and biomass. The metabolic engineering of the Arg catabolism pathway towards inducing TAGs accumulation without the need for -N or any stress-induced condition may be the next necessary breakthrough for expanding microalgae industrial applications.

## Figures and Tables

**Figure 1 cells-12-01379-f001:**
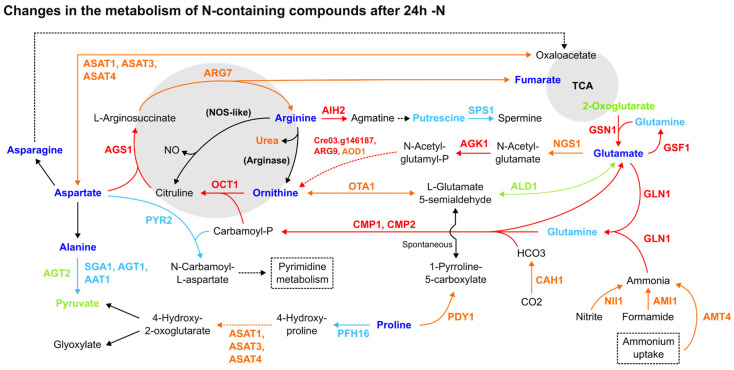
Comparative summary of the changes in the Arg metabolism and adjacent pathways after 24 h of -N. Colour meaning of the metabolites, enzymes and arrows: Black: not detected or not significant; Red: upregulated in 2 or more datasets; Orange: upregulated in one dataset, not detected in the others; Dark blue: downregulated in 2 or more reports; Light blue: downregulated in one report, not detected in the others; Light green: detected, without agreement among datasets about its up- or downregulation. Enzyme name abbreviations: AAT1: alanine aminotransferase (Cre10.g451950); AGK1: acetylglutamate kinase (Cre01.g015000); AGS1: argininosuccinate synthase (Cre09.g416050); AGT1: alanine-glyoxylate transaminase (Cre06.g294650); AGT2: alanine-glyoxylate transaminase (Cre03.g182800); AIH2: agmatine iminohydrolase (Cre01.g009350); ALD1: aldehyde dehydrogenase (Cre12.g520350); AMI1: formamidase/acetamidase (Cre16.g691750); AMT4: ammonium transporter (Cre13.g569850); AOD1: acetylornithine deacetylase (Cre02.g105500); ASAT3: aspartate aminotransferase (Cre02.g097900); ASAT4: aspartate aminotransferase (Cre06.g257950); ARG7: argininosuccinate lyase (Cre01.g021251); ARG9: acetylornithine aminotransferase (Cre06.g278163); CAH1: carbonic anhydrase (Cre04.g223100); CMP2: carbamoyl phosphate synthase, small subunit (Cre06.g308500); GLN1: glutamine synthetase (Cre02.g113200); GSF1: ferredoxin-dependent glutamate synthase (Cre12.g514050); GSN1: glutamate synthase, NADH-dependent (Cre13.g592200); NGS1: (Cre16.g694850); NII1: nitrite reductase (Cre09.g410750); OCT1: ornithine carbamoyltransferase (Cre12.g489700); OTA1: ornithine transaminase (Cre11.g474800); PFH16: prolyl 4-hydroxylase 16 (Cre08.g369300); PDY1: proline dehydrogenase (Cre01.g036850); PYR2: aspartate carbamoyltransferase (Cre02.g079700); SGA1: serine glyoxylate aminotransferase (Cre01.g005150); SPS1: spermine synthase (Cre06.g251500); Cre03.g146187: N-acetyl-gamma-glutamyl-phosphate reductase/NAGSA dehydrogenase. Enzymes between parentheses, in bold: not found in *C. reinhardtii* genome. Dashed lines: simplified pathways.

**Figure 2 cells-12-01379-f002:**
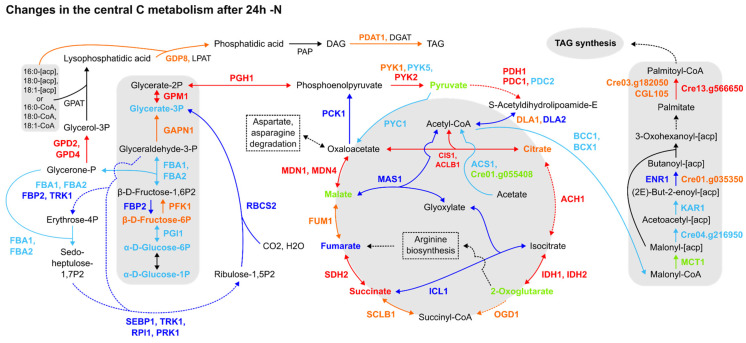
Comparative summary of the changes in the Calvin cycle, glycolysis/gluconeogenesis, TCA cycle, fatty acid synthesis and TAG synthesis pathways after 24 h of N deprivation. Colour meaning of the metabolites, enzymes and arrows: Black: not detected or not significant; Red: upregulated in 2 or more datasets; Orange: upregulated in one dataset, not detected in the others; Dark blue: downregulated in 2 or more papers; Light blue: downregulated in one paper, not detected in the others; Light green: detected, but no agreement among datasets about its up- or downregulation. Enzyme name abbreviations: ACH1: aconitate hydratase (Cre01.g042750); ACS1: acetyl-CoA synthetase/ligase (Cre01.g071662); BCC1: acetyl-CoA biotin carboxyl carrier (Cre17.g715250); BCX1: acetyl-CoA carboxylase beta-carboxyltransferase subunit beta (Cre12.g484000); CIS1: citrate synthase, mitochondrial (Cre12.g514750); CGL105: predicted protein (Cre12.g507400); DLA2: dihydrolipoamide acetyltransferase, possibly plastid (Cre03.g158900); DGAT1: diacylglycerol acyltransferase (Cre01.g045903); ENR1: enoyl-[acyl-carrier protein] reductase (Cre06.g294950); FBA1: fructose-1,6-bisphosphate aldolase (Cre01.g006950); FBA2: fructose-1,6-bisphosphate aldolase (Cre02.g093450); FBP2: fructose-1,6-bisphosphatase (Cre12.g510650); FUM1: fumarate hydratase (Cre06.g254400); GAPN1: glyceraldehyde 3-phosphate dehydrogenase, non-phosphorylating (Cre12.g556600); GDP8, glycerophosphoryl diester phosphodiesterase family protein (Cre01.g000300); GPAT: glycerol-3-phosphate phosphate acyltransferase, contains PlsC domain (Cre06.g273250); GPD2: glycerol-3-phosphate dehydrogenase, dihydroxyacetone-3-phosphate reductase (Cre01.g053000); GPD4: glycerol-3-phosphate dehydrogenase, dihydroxyacetone-3-phosphate reductase (Cre10.g421700); GPM1: phosphoglycerate mutase, 2,3-bisphosphoglycerate-independent (Cre06.g272050); ICL1: isocitrate lyase (Cre06.g282800); IDH1: isocitrate dehydrogenase, NAD-dependent (Cre17.g728800); IDH2: isocitrate dehydrogenase, NAD-dependent (Cre02.g143250); KAR1: 3-oxoacyl-[acyl-carrier protein] reductase (Cre03.g172000); LPAT: 1-Acyl-sn-glycerol-3-phosphate acyltransferase (Cre17.g707300); LPAT1: 1-Acyl-sn-glycerol-3-phosphate acyltransferase (Cre09.g398289); MAS1: malate synthase (Cre03.g144807); MCT1: malonyl-CoA:acyl-carrier-protein transacylase (Cre14.g621650); MDN1: NAD-dependent malate dehydrogenase, chloroplastic (Cre03.g194850); MDN4: malate dehydrogenase 4 (Cre12.g483950); OGD1: 2-oxoglutarate dehydrogenase, E1 subunit; (Cre12.g537200); PCK1: phosphoenolpyruvate carboxykinase (Cre02.g141400); PDAT1: lecithin:cholesterol acyltransferase (Cre02.g106400); PDC1: mitochondrial pyruvate dehydrogenase complex, E1 component, alpha subunit (Cre07.g337650); PDC2: pyruvate dehydrogenase, E1 component, alpha subunit (Cre02.g099850); PDH1: pyruvate dehydrogenase E1 beta subunit (Cre16.g677026); PFK1: phosphofructokinase (Cre06.g262900); PGH1: enolase (Cre12.g513200); PGI1: phosphoglucose isomerase (Cre03.g175400); PRK1: phosphoribulokinase, chloroplast precursor (Cre12.g554800); PYC1: pyruvate carboxylase (Cre06.g258700); PYK1: pyruvate kinase (Cre12.g533550); PYK2: pyruvate kinase (Cre06.g280950); PYK5: pyruvate kinase (Cre02.g147900); RBCS2: ribulose-1,5-bisphosphate carboxylase/oxygenase small subunit 2 (Cre02.g120150); RPI1: ribose-5-phosphate isomerase (Cre03.g187450); SCLB1: succinate-coa ligase beta chain (Cre17.g703700); SDH2: iron-sulphur subunit of mitochondrial succinate dehydrogenase (Cre06.g264200); SEBP1: sedoheptulose-1,7-bisphosphatase (Cre03.g185550); TRK1: transketolase (Cre02.g080200); PAP1: phosphatidate phosphatase (Cre05.g230900); PAP2: phosphatidate phosphatase (Cre05.g240000); Cre01.g035350: mitochondrial trans-2-enoyl-CoA reductase (MECR, NRBF1); Cre01.g055408: acetyl-CoA synthetase (ACSS, acs); Cre03.g182050: protein ACS-13, isoform C; Cre04.g216950: beta-ketoacyl-[acyl-carrier-protein] synthase III/KASIII; Cre13.g566650: long-chain acyl-CoA synthetase 2. Dashed lines: simplified pathways.

**Figure 3 cells-12-01379-f003:**
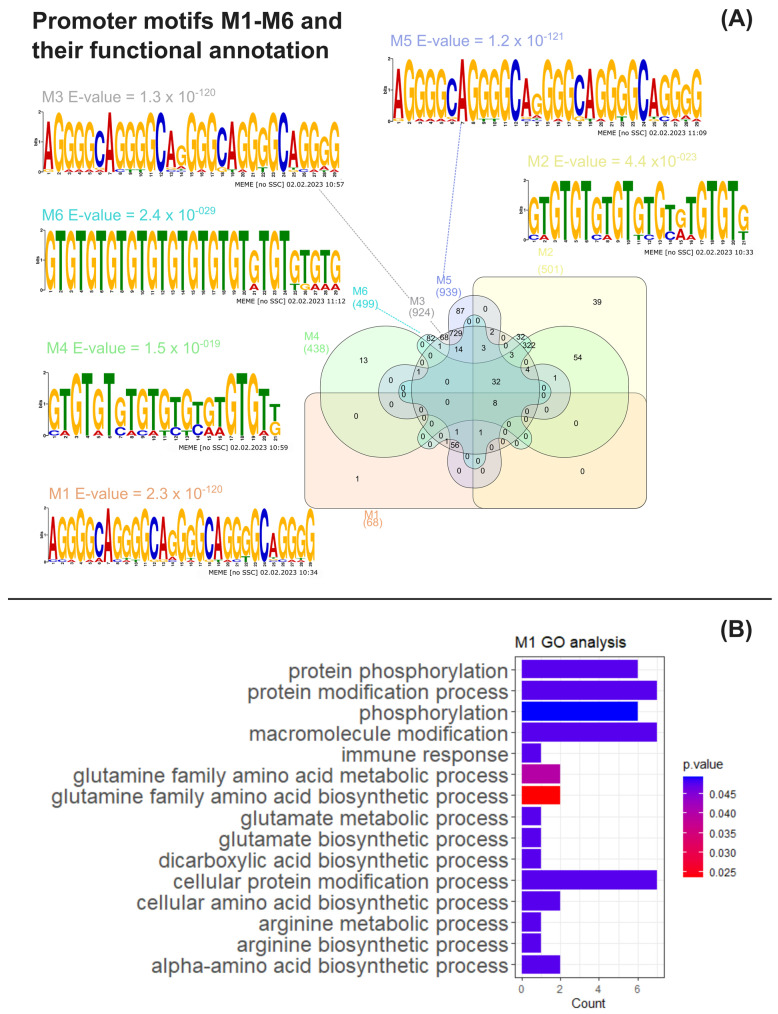
MAST motif analysis and functional analysis of MAST gene clusters. (**A**) Venn diagram obtained using the InteractiVenn tool of the number of *C. reinhardtii* genes whose common putative conserved promoter motifs M1–M6, together with the weighted matrix logos of M1–M6 and motif E-values obtained from MEME analysis; (**B**) significant overrepresented Gene Ontology terms for biological process found by BinGO analysis associated with the M1 gene cluster obtained through MAST analysis [33].

## Data Availability

Datasets for this research are included in Blaby et al. [1], Boyle et al. [2], Goodenough et al. [9], Park et al. [5], Valledor et al. [7] and Wase et al. [8].

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
