# Peer review of "From Feasting to Fasting: The Arginine Pathway as a Metabolic Switch in Nitrogen-Deprived *Chlamydomonas reinhardtii"

_cells, 2023, doi:10.3390/cells12101379_

Round 1
Reviewer 1 Report
I would like to congratulate the authors for their paper. it was well built and the presented results support its conclusions. The chosen subject is of great interest and is likely to attract the attention of different readers. Therefore, I am in favor of publishing this manuscript.
Reviewer 2 Report
Chlamydomonas reinhardtii is one of the most promising species for use in food and feed production. The presented manuscript is based on the analysis of previously obtained data. The authors successfully addressed the major metabolic pathways involved in response, uptake, and exploration by reanalyzing omics data from previously published datasets. The study draws on a large number of sources. References are dominated by contemporary publications. However, the structure of the manuscript lacks a conclusion. The discussion ends, and the conclusions remain hidden in the text. The introduction, on the contrary, ends with the results obtained instead of setting the goal of the work and formulating the hypothesis. It seems to me that structuring according to the classical scheme of a scientific article will improve the manuscript.
Reviewer 3 Report
The authors analysed the effects of nitrogen deprivation in Chlamidomonas reinhardtii through the reanalysis of omics data from 6 previously published datasets.
General comments
This is an interesting article, concerning microalgae metabolism, in particular under nitrogen starvation. The hypothesis is clearly stated and material and methods clearly reported and detailed. The results obtained after reanalysis of omics data previuosly published are confirmatory of previous findings and offer new insight and advancement of knowledge in the field.
However, it is difficult to read because of the large amount of the data analyzed and reported. In particular, the discussion is presented in a not well-stuctured manner and apart from the first paragraph, is quite confusing and sometimes not focused on the results. For example, the last paragraph discusses the role of NO, which has not been analyzed by the authors. It is suggested that discussions be reorganized, taking into consideration the results obtained and focused on the main evidence found. In this regard, a table summarizing the most important findings might be helpful. Also sometimes the sentences are disconnected from each other (e.g. lines 404-411).
Moreover, a general conclusion summarising all the relevant findings ad suggestions for future research is lacking and should be added. As an example, the sentence at lines 357-360 could be moved to the conclusion section.
Specific comments
Abstract –
line 21: I suppose “unraveling” instead”unrevealing”
Introduction –
line 82: suggests;
lines 84-87: this is a complex sentence; to split it in two separate sencences coulld help the reader.
Material and methods –
lines 98-101: which strains were used by Park et al., Valledor et al., and Wase et al.? This important information is lacking.
Lines 128: on which basis was the length of 1500 bp upstream chosen?
Line 133: was instead of were.
Results –
Line 159: “arginine, asparagine…were downregulated”. Downregulation id more appropriate for the regulation of gene expression, in this case “decreased” is more appropriate.
Lines 232-235: the sentence is not clear, because ppalmitoil-CoA biosynthesis starts from acetyl-CoA which is carboxylated to malonyl-CoA.
Line 291: oxidyzing instead of oxidative.
Line 294: reducing instead of reductive.
Discussion – (see the general comment above)
Lines 385-386: probably because some aminoacids are used for acetyl-CoA synthesis?
Line 388: to what is referred “this”?
Lines 405-406: this sentence does not connect with what was said above.
Lines 445-447: it is not clear why the negative control of PII on Acetyl-CoA carboxilase (the first enzyme of the biosynthetic pathway of palmitate) leads to higher TAG yelds.
Reviewer 4 Report
Dear authors,
In this study, the authors use different strategies to investigate the factors on which the production of lipids in Chlamydomonas depends, finding that amino acids and especially arginine play a key role.
The paper is well written, and the experiments carried out in an adequate way. The strategy used is quite innovative and the data obtained is valuable for experts in algae and especially Chlamydomonas. However, I see a series of limitations that must be clearly stated, such as the case of NIT2 and its role in this study, which must be answered and indicated before its publication.
Majors:
*Introduction: In Chlamydomonas the gametogenesis is also induced and produced in a medium without nitrogen, cite and discuss this fact, please.
*A key point for the interpretation of the results is the genetic background of the strains of chlamydomonas used, in the materials and methods there should be a section in this regard in which it is clearly indicated. And above all, are they any mutants in the main nitrogen transcription factor in Chlamydomonas NIT2? I think they are all mutants in NIT2.
*Figure 1: Missing nitrate reductase and nitrate and nitrite transporters
*Arginine is the only amino acid that in Chlamydomonas can be transported to the interior of the cell and used as a source of nitrogen, so there must be transporters, include them in Figure 1, please.
*L205: Indeed, LAO1 is a fundamental enzyme. Cite and discuss the latest work on LAO1 in Chlamydomonas 10.1016/j.algal.2018.101395
*L447: In the entire text, NIT2 is only cited here, when it is the master regulator gene in nitrogen assimilation. The lack of contextualization of the data with this gene was notably lacking. Please discuss the data with more in mind the role of NIT2 please.
*Clearly state and discuss the consequences of under which phototrophic, heterotrophic, or mixotrophic conditions the experiments were performed. Discuss the implications, especially with regard to the metabolism of acetate.
*In the discussion the study of the promoters carried out is practically not dealt with, please discuss these data and contextualize them with the others.
*Please, do not end the discussion in this way, at the end of the discussion it is also always convenient to make a brief summary of the most important thing found.
*It would also be convenient in the discussion to propose how these studies can biotechnologically help the use of Chlamydomonas as a producer of TGA or biomass.
Minors:
L76: “nitrogen deprivation (-N)”. Previously already mentioned in line 48, 123
L105: “DE” define please
Round 2
Reviewer 3 Report
Dear Authirs,
all previous suggestions have been accepted.
I suggest a careful check of text editing. Below are some examples:
line 439: suggests
line 440-441: the sentence is not clear
line482: Nitric oxide, NO, nitric oxide: please uniform
line 500: the known master regulator
lines 532-533: to anticipate rapidly the acquisition and ssimilation.
Finally, in the reviewers' opinion, the discussion is still too long and confusing and should be improved.
Author Response
We thank the reviewer for the valuable comments and time, and we hope to have addressed all the reviewer’s concerns.
I suggest a careful check of text editing. Below are some examples:
line 439: suggests
RES: We modified the text accordingly
line 440-441: the sentence is not clear
RES: The sentence was modified
line482: Nitric oxide, NO, nitric oxide: please uniform
RES: The text was revised accordingly
line 500: the known master regulator
RES: The text was revised accordingly
lines 532-533: to anticipate rapidly the acquisition and ssimilation.
RES: The text in conclusion remarks was revised accordingly
Finally, in the reviewers' opinion, the discussion is still too long and confusing and should be improved.
RES: The discussion session was modified, separating different sessions focusing on the main aspects of each topic of information, reducing the overload of shifts on discussions about omics data and promoter analysis
Reviewer 4 Report
I think the authors have responded positively to most of my suggestions, and I approve of the paper in its current version.
Author Response
We thank the reviewer for your comments and hope to have addressed all concerns.